# The Floral Signals of the Inconspicuous Orchid *Malaxis monophyllos*: How to Lure Small Pollinators in an Abundant Environment

**DOI:** 10.3390/biology11050640

**Published:** 2022-04-21

**Authors:** Edyta Jermakowicz, Joanna Leśniewska, Marcin Stocki, Aleksandra M. Naczk, Agata Kostro-Ambroziak, Artur Pliszko

**Affiliations:** 1Department of Plant Biology and Ecology, Faculty of Biology, University of Bialystok, Ciołkowskiego 1J, 15-245 Bialystok, Poland; joanle@wp.pl; 2Institute of Forest Sciences, Faculty of Civil Engineering and Environmental Sciences, Bialystok University of Technology, Wiejska 45E, 15-351 Bialystok, Poland; m.stocki@pb.edu.pl; 3Department of Evolutionary Genetics and Biosystematics, Faculty of Biology, University of Gdańsk, Wita Stwosza 59, 80-308 Gdańsk, Poland; aleksandra.naczk@ug.edu.pl; 4Laboratory of Insects Evolutionary Biology and Ecology, Department of Genetic and Zoology, Faculty of Biology, University of Bialystok, Ciołkowskiego 1J, 15-245 Bialystok, Poland; ambro@uwb.edu.pl; 5Institute of Botany, Jagiellonian University, Gronostajowa 3, 30-387 Krakow, Poland; artur.pliszko@uj.edu.pl

**Keywords:** flower anatomy structure, GC-MS, SEM, SPME, TEM, visiting insects, volatiles

## Abstract

**Simple Summary:**

*Malaxis monophyllos* is an ephemeral orchid with very small, greenish flowers, whose pollination system remains vague. Therefore, the authors aimed to identify the flower’s features, including its anatomical micro and ultrastructure as well as scent biochemistry, which are involved in attracting pollinators. In this paper, the authors established the variety of emissions of *M. monophyllos* volatile compounds, with a high proportion of aldehydes and aliphatic alcohols, listed as chemicals that induce a pronounced reaction in Diptera. Second, the entire *M. monophyllos* labellum exhibits metabolic and secretory activity, which can be related to both emission of volatiles and visual attractants but also to the nutritional reward for pollinators. All these flower features indicated that its pollination system is dedicated to dipterans, with few signaling modalities corresponding to deceptive species (brood site and food decoy) but also referring to rewarding ones (nutritional secretion, fungus/microbe reward). This research reveals a few new issues in *M. monophyllos* pollination biology that provides new scientific areas for in-depth insights in the future.

**Abstract:**

Many orchid species have evolved complex floral signals to ensure pollination efficiency. Here, the authors combined analyses of anatomical flower structure with analyses of the volatile composition and flower-visiting insects’ behaviour, as well as characterised features that can attract pollinators of the inconspicuous orchid *Malaxis monophyllos*. During field observations, the authors found that only small Diptera (e.g., mosquitos, drosophilids, fungus gnats) visit and are interested in the flowers of *M. monophyllos*, which was reflected in the characterised flower features that combine well with the pollination system, which engages dipterans. Analyses of the *M. monophyllos* floral scent revealed substantial concentrations of aliphatic compounds, e.g., 1-octen-3-ol and 1-octanol, which condition the mushroom-like scent and a substantial fraction of alkanes, some of which have been previously described as sex mimicry and aggregation pheromones in orchids’ deceptive systems. The labellum anatomical structure exhibits a highly diverse cell cuticle surface and pronounced metabolic and secretory activity of the epidermal and subepidermal cells from all parts of the labellum. Moreover, our study provides evidence for the subsequent decoys of *M. monophyllos* flowers, including visual signals, such as raphides located on the labellum margin and the rewarding ones connected with lipid secretion limited to the area behind the column. Taking an integrative approach to studying *M. monophyllos* pollination biology, the authors provide new insight into its previously vague pollination strategies and provide evidence for complex floral signal operation in luring potential pollinators. The synergistic effect of *M. monophyllos* flowers’ volatile and visual signals, together with additional rewarding for nectar/fungus/microbe-feeding pollinators, requires further detailed investigation that will be invaluable in explaining the evolution of Diptera-specific pollination systems in orchids.

## 1. Introduction

The flowers of both deceptive and rewarding orchids must emit signals that stimulate targeted insects, and, in particular, cause the involvement of pollinators [1,2,3,4,5,6]. In their usually short life spans, potential pollinators try to feed, mate, and breed as effectively as possible [7]. Further, they often appear briefly, and their cohorts may change during the period of plant anthesis, thus establishing a wide spectrum of chemical and physical floral attractants create an intricate but more effective pollination system. The efficient detection and processing of such multimodal attractants by one group or luring of different functional groups of pollinators (exerting different selection pressures) increases the likelihood of pollination [8,9]. This can be particularly adaptive in changing environments during prolonged orchid flowering and may point towards the bet-hedging strategy, which allows plants to be pollinated even if the preferred pollinators are lacking [10,11,12,13].

The pollinators’ detectability of flowers depends on many factors, especially connected with a floral display, associated with flower number and height of inflorescence [14,15,16] but also with flower structure and colour [17]. However, the chemical attractants (fragrance composition) are one of the most relevant and often the most complex floral signals. The enhanced fragrance analysis techniques make the plant’s pollination systems much more complicated than previously thought ([13,18] for review). For instance, Braunschmid and Dötterl [19] found several scent phenotypes in two studied populations of *Cypripedium calceolus*, and Wróblewska et al. [20] described various volatile compounds (pheromones, substances of social and feeding insects’ behaviour) secreted by deceptive *Dactylorhiza* species, pointing to the complex character of its pollination systems.

Great temporal differences in fragrance composition have also been identified for two species of rewarding *Gymnadenia*, which entices different functional groups of insects [21]. Similarly, in the case of a few deceptive species (*Epidendrum ciliare*, *Anacamptis morio*, *Orchis mascula*, and *O. pauciflora*), high individual variability in floral fragrance has also been observed, which attract different groups of visiting insects and, in turn, condition the complex pollination system of these species [22,23,24].

The benefits of combinations of different pollination strategies can be particularly important in the case of unremarkable, small species whose visual and volatile traces might be strongly spatially limited. Such species are common in the genus *Malaxis* Sw. (Malaxidinae, Orchidaceae), which includes about 300 species, found mostly throughout the tropics and subtropics of the Old and New Worlds [25]. Most of them have greenish, very small (3 to 10 mm in span), inconspicuous flowers, which makes their pollination biology mostly unexplored. Nonetheless, they are intriguing and important in understanding the evolution of different pollination strategies in orchids. In general, less conspicuous visual displays in orchids, often supported by particular scent bouquets, are associated with pollination by sexual deception or brood-site mimicry [26,27,28]. Although some observations indicate the presence of secretions on the lips of some *Malaxis* species, the rewardless strategy is more often emphasised in the case of different members of this genus [29]. Other reports pointed out strange and disagreeable odours as the most common in the tribe [30,31,32], but with at least one pleasant exception of the *Malaxis rzedowskiana* scent, which was reminiscent of violets [33]. The flora of Europe has one species, *Malaxis monophyllos* (L.) Sw., which has one of the smallest flowers among European orchids. It is associated with slightly shady and moderately humid environments and is medium-rich in competitive plant species [34]. The populations of *M. monophyllos* are usually small, often highly dispersed, and stand out with huge fluctuations in the number of individuals between years; thus, effectively attracting pollinators and successful pollination might be a challenge.

Furthermore, demographic surveys of *M. monophyllos* populations revealed that their size and dynamics are greatly influenced by the process of population renewal, and therefore depend indirectly on the potential for generative reproduction [34], which was emphasised by the low to moderate fruit set (from 1.5% to about 40%) observed in *M. monophyllos* populations [31,34,35]. In the context of pollination and mating mode, such outcomes can be explained by a non-rewarding, self-incompatible, pollinator-dependent system [3,36,37]. Although, there are no detailed studies regarding *M. monophyllos* pollination biology, and the cursory information is based only on scarce pollinator observations and estimating fruiting levels [31,35,36]. Up to now, the widest observations of visiting insects and pollinators were documented by Claessens and Kleynen [31] in the alpine *M. monophyllos* populations, which confirmed the earlier notes of Vöth [38] concerning fungus gnats from the Mycetophilidae family (Diptera) as the main group of successful pollinators. Flies are frequently mentioned as the main visitors of different species of genus *Malaxis* [29]. Myophily is widespread in orchids, and flies are specified as important pollinators in both rewarding and deceptive pollination systems [39,40,41]. Among them are nectar-, fungus-, and yeast-feeding *Drosophilideae* [42,43], which might also be misled in this regard [7]. Orchids might also deceive flies through sexual attractants or brood site mimicry [39,40,44]. In *M. monophyllos*, flies were supposedly attracted by a faint, mushroomy, musk-like scent [31,37] or scents conjuring crustacean and algae associations [45] that imitated food or brood site resources for them, although these statements have not been confirmed so far.

Thus, the presented surveys aimed to characterise and understand the *Malaxis monophyllos* floral traits that are involved in attracting potential pollinators. For this purpose, the authors used the combining analyses of the flowers’ anatomical structure, including micro and ultrastructure, together with volatiles’ composition and behavioural observations of insects visiting *M. monophyllos* flowers.

## 2. Study Species and Population

*Malaxis monophyllos* produces a dozen to several dozen yellow-greenish flowers per racemose inflorescence, which open gradually from the bottom upward (Figure 1A). The pedicel of the ovary is twisted through 360 degrees (due to twisting of the flower stalk), which makes the lip point upwards. The median sepal points down and the petals extend out, improving the landing zone for visiting insects [31]. The two pairs of pollinia are referred to as hard and waxy [46]. Flowers on the inflorescence open gradually from the bottom upwards, over about two weeks (depends on flower number). Moreover, different shoots may start to bloom at different times, and thus the flowering can be additionally extended, starting in the first part of June and ending in the middle of July (depending on weather conditions). All observations and plant material collection were performed in a population located in Wigry National Park (north-eastern Poland) in June–July of 2014, 2017, and 2019. Wigry National Park lies in the temperate climate zone, where the mean annual air temperature is about 6.5 °C and the mean annual precipitation is about 600 mm. The shoots of *M. monophyllos* are scattered on the edge of a spruce forest (*Sphagno girgensohnii-Piceetum*) with a rich moss layer and in a moderately to highly humid habitat. In 2014, the shoots of *M. monophyllos* were scattered over an area of 0.2 ha, where the authors found 43 shoots and, therein, 20 flowering shoots. In 2016, the authors extended the monitored area to about 0.5 ha, thus in the years 2017 and 2019, the population consisted of 155 and 145 shoots respectively, with about 30% of flowering shoots in both years.

## 3. Plant Material

### 3.1. Plant Material Collection

The flowers of *M. monophyllos* were collected in 2017. Due to the high conservation status of species in Poland, the collection took place in accordance with the permit issued by the Polish Ministry of Environment (Jermakowicz; OP-WPN.286.111.2017.MD). The collection was performed in different stages of plant anthesis: (1) at the beginning—up to three days from opening, and (2) after the optimum of anthesis—about seven days after opening. In total, the authors collected twenty flowers from four inflorescences. The collected flowers were stored in the Faculty of Biology University of Bialystok in Poland.

### 3.2. Anatomical Analyses of Flowers (LM, TEM, SEM)

The freshly collected flowers were tentatively observed under a stereomicroscope to identify structurally distinct areas in the flowers, as well as glandular flower structures. Freshly opened flowers (immersed in water with a little of Tween) were viewed with a Nikon Eclipse E600 fluorescence microscope equipped with filter UV-2A (EX 330-380, DM 400, BA 420) to image the places of intensive autofluorescence, which are visible under UV light.

The flowers for anatomical investigation in LM (light microscopy) were conserved in FAA (formaldehyde-alcohol-glacial acetic acid) for 24 h [47], rinsed with 70% ethanol 3 times, and stored in 70% ethanol at RT (room temperature). Before staining, the plant material was rehydrated. Whole fixed flowers were stained for several min with 1% Safranin O in 50% ethanol with Alcian blue solution mixed in a 1:1 ratio, then washed in water and mounted in 50% glycerol. The solution of Trypane blue (0.25 g Trypane blue, 250 mL glycerol, 125 mL distilled water, and 125 mL lactic acid (85%)) was used to visualize fungi inside and on the surface of the flower; whole flowers were immersed in a drop of this solution for several minutes, washed in water, and mounted in 50% glycerol [48] with modifications]. Moreover, semisections of labellum material were treated with Sudan Black B (SBB) to detect the presence of lipids [49]. Prepared material was then analysed for general histology, as well as for detailed structures using an Olympus BX53 light microscope equipped with the Olympus Digital Colour Camera SC30 and Olympus CellSens Standard imaging software (version 1.11).

For scanning electron microscopy (SEM), after dehydration in an ethanol series, the samples were dried by the critical point method using liquid CO_2_ coated with gold. They were then examined using a Philips XL-30 SEM at an accelerating voltage of 15–20 kV (Laboratory of Electron Microscopy, University of Gdańsk, Poland). To complete the picture, the few samples underwent a separate procedure with post-fixation in 1% OsO_4_ in bidistilled water during 6 h. After dehydration in the ethanol series and draining in the critical point, samples were mounted to an aluminium stub and coated with a gold alloy. Then they were viewed with a scanning electron microscope Zeiss Leo 1430VP (Imaging Laboratory, University of Warsaw, Poland) (Figure 2F and Figures 6F and 7A,D). For ultrastructural analysis using transmission electron microscopy (TEM), the floral material was fixed in 2.5% (*v*/*v*) glutaraldehyde (GA) and 2.5% (*w*/*v*) paraformaldehyde (PFA) in a 0.05 M cacodylate buffer (pH = 7.0). The material was then post-fixed overnight in 1% (*w*/*v*) OsO_4_ in a cacodylate buffer in a refrigerator and rinsed in the fresh buffer. After 1 h in a 1% (*w*/*v*) aqueous solution of uranyl acetate, the material was dehydrated with acetone and embedded in Spurr’s resin. Ultrathin sections were cut on a Leica EM UC 7 ultramicrotome with a diamond knife and stained with uranyl acetate and lead citrate. The sections were examined using a Tecnai G2 Spirit BioTwin FEI transmission electron microscope (Laboratory of Electron Microscopy, University of Gdańsk, Poland) at an accelerating voltage of 120 kV. For the above studies, samples were prepared in accordance with previously described procedures (e.g., [50,51]).

### 3.3. Fragrance Collection and Analysis

Two volatile samples, which were the entire *M. monophyllos* inflorescence (cut off just below the first flower), containing about 25–30 opened flowers each, were enclosed in a headspace vial and transported to the Laboratory of Chemistry at the Faculty of Forestry, Bialystok University of Technology (Hajnówka, Poland). The volatile samples were analysed using headspace solid-phase microextraction and gas chromatography-mass spectrometry (HS-SPME/GC-MS) methods. The research was performed using previously developed methodology [52,53].

The entire inflorescence of *M. monophyllos* (5 g) was placed in a 60 mL headspace vial and heated for 30 min at 40 °C. For research, the fibre with the divinylbenzene/carboxen/polydimethylsiloxane (DVB/CAR/PDMS) stationary phase was used, as the DVB/CAR/PDMS fibre provides for effective extraction of volatile samples characterized by different polarities [54,55].

After 30 min of exposition, the SPME fibre was introduced for 10 min into the injection port of the GC–MS apparatus. The GC–MS analyses were conducted using an Agilent 7890A gas chromatograph with an Agilent 5975C mass spectrometer. Chromatographic separation was performed on an HP-5MS capillary column (30 m × 0.25 mm × 0.25 μm) at a helium flow rate of 1 mL/min. The injector worked in a splitless mode at a temperature of 250 °C. The initial column temperature was 35 °C and rose to 250 °C at 5 °C/min. The ion source and quadrupole temperatures were 230 °C and 150 °C, respectively. Electron ionization mass spectrums (EIMS) were obtained at ionization energy of 70 eV. The detection was performed in a full scan mode from 29 to 600 a.m.u.

After integration, the percentage content of each component in the total ion current (% TIC) was calculated. To identify components, both mass spectral data and the calculated retention indices were used. Mass spectrometric identification was performed with an automatic system of GC–MS.

### 3.4. The Community of Flower Visitor Insects

The preliminary insect observations of *M. monophyllos* flower visitors in 2014 year were conducted during the flowering peak (in the first part of June), two times a day, from 9:00 to 12:00 am and from 2:00 to 6:30 pm (over 4 days). These observations indicated a complete lack of insect activity on *M. monophyllos* flowers during the first part of the day and the early afternoon. The activity of visiting insects started around 6:00 pm. Thus, the authors repeated observations in 2019 year, from 5:00 pm to 8:30 pm. Additionally, the previous long-term demographic observation (2008–2014) indicated that the highest fruiting was recorded at the bottom of flowering stalks. This, in turn, may suggest that the highest pollinator abundance is at the beginning of the anthesis. Thus, the observations of *M. monophyllos* flower visitor insects were performed on the eight flowering individuals each day, during four rainy but warm days in the second part of June. The selected flowering stalks were video recorded using digital cameras (HDR-PJ780 Sony Corp. Japan) (with the possibility of night time recording) (three inflorescences) and directly observed (from one to three inflorescences) each day. In total, 42 h of recording and observation were analysed. Some visiting insects were captured using an aspirator, determined (to orders, families, or species) and categorized into visiting or pollinating insect groups. Insects were regarded as pollinators when they carried on the *M. monophyllos* pollinaria. Flower visitors landed on *M. monophyllos* flowers but did not carry the pollinaria. When visitors were not captured, their number and taxa were recorded with achievable accuracy. The authors reported the number and behaviour of all recorded insects and assigned, for each specimen, the mean number of flowers per inflorescence visited during the single bout (N_f_) and the flower visitation rate, established as the mean number of flowers visited per second (T_f_) [56].

## 4. Results

### 4.1. Anatomy, Morphology, and Ultrastructure of M. monophyllos Flowers

Fresh flowers of *Malaxis monophyllos* are approximately 3 mm long (Figure 1B). The pointing-upwards labellum is concave and has folded margins, which makes the column slightly hidden inside them. The column structure, particularly the clinandrium-rostellum arrangement and deep embedding of pollinia in pollen pockets, prevents spontaneous autogamy (Figure 1C,D). The adaxial surface of the labellum has a few micromorphologically characteristic areas, due mostly to cuticle structure and secretory properties, which indicated the direct observation supported by the SEM (Figure 2A). The innermost, lover part of the lip, right behind the column (Figure 2A,B,E,F) is smooth and shiny. This area is covered with a very thick, continuous and tight cuticle layer, extending independently of the cellular pattern (Figure 2E,F). In some parts of this covering, the perforations are visible as well as the residues of secretory substances (Figure 2E,G). Moreover, in epidermal and subepidermal cells of the innermost part of the labellum, the authors found greater accumulation of lipids (Figure 2H). The cells bordering this inner area is characterised by a rough structure (Figure 2D,G), caused by the multiple branched outgrowths that are formed by cell wall of these cells (well visible in Figure 3A–D) additionally covered by strongly wrinkled cuticles. The surface of such a formed cell wall is huge, which could be related to the increasing evaporation area. The numerous crypts and cavities of these cell walls are the places where the various homogenous and granular secreted substances were observed (Figure 3D–G). The dense parietal cytoplasm of these cells with numerous mitochondria, extended membrane system (both rER and sER profiles), dictyosomes, various vesicles, myelin-like structures, ribosomes, and plastids with plastoglobuli indicate and confirm secretory function of these cells (Figure 3D–L). Additionally, indication is also present of the vesicles fusing with the plasma membrane and released their content beyond the protoplast (Figure 3H), as well as groups of vesicles that released towards the cell wall (Figure 3I).

The folded margins of the bottom part of the labellum envelop a huge aggregation of papillary cells and trichomes, probably functioning as osmophores (Figure 2B, marked as 3, Figure 4A,B). Papillae are single vacuolated (convex to elongated and finger-like) epidermal cells with lamellar outgrowths of the outer cell wall, covered with cuticle (Figure 4C–E). The residues of a flocculent and granular secretory substance were also observed on its surface (Figure 4E). TEM observations showed a well-developed membrane system (mainly rER profiles), abundant mitochondria, and plastids gathered around a large nucleus containing plastoglobulins (osmiophillic globules) (Figure 4C–E). Two types of trichomes were observed: unicellular (elongated, finger-like) and bicellular (with long stalk cell and head cell), which differ in terms of cuticle structure (Figure 4B,G,H). All cells of trichomes are highly vacuolated (Figure 4F–H) and contain a large number of expanded rER cisternae, numerous vesicles of various size, ribosomes, and plastids with plastoglobuli (Figure 4G,H). In a bicellular trichome, the grainy contents of ER cisternae are released towards the cell wall, which is unevenly thickened and supersaturated with a huge amount of osmophilic material transported to the cell surface (Figure 4I). Apart from the lip, the authors found single, three-cell trichomes, rarely scattered on the petals and sepals (Figure 4J,K). Their function is unclear.

The most central part of *M. monophyllos* labellum (Figure 2C—marked as 5), is covered by cells with different cuticle striation pattern (Figure 5A–C). The ultrastructure analyses indicated different thickness of its cell walls (even of a single cell) with a variety of outgrowths covered by cuticle with distinctive microchannels (Figure 5D–K). The residues of secretory substances, both homogeneous and granular, were present on the surface (Figure 5G). The protoplast of these epidermal cells is typical of metabolically active cells, thus it contains numerous mitochondria with crista, extended membrane system (both rER and sER profiles), dictyosomes, various vesicles, myelin-like structures and ribosomes (Figure 5F–L). In epidermal cells the plastids with plastoglobuli were observed, while in subepidermal cells both plastids with starch grains and with plastoglobuli, as well vacuoles with tannin-like materials were detected (Figure 5K,L).

The marginal part of the labellum has a distinctive, bulging structure (Figure 6A) that show luminescence in the UV light (Figure 6B). In this part of the labellum, in transparent, chemically fixed flowers, the authors found a huge aggregation of idioblasts with raphides (Figure 6C,D), while in other parts of the flower, the cells with raphids were rare (not shown on figures). The idioblasts are grouped by several only in these bulges mostly in the subepidermal cells (Figure 6D). In some places however, the cuticle is smooth (Figure 6E). The further analyses revealed that some of an idioblasts have a wall emerging from between the epidermal cells (Figure 6H–J). Moreover, in the case of some idioblasts, the authors observed that under the pressure of a mechanical stimulus (e.g., touch of preparation tools), the raphides perforated the wall and were ejected from the idioblasts outside the cells (Figure 6D,F). Most idioblasts do not eject the raphides.

Almost all observed flowers of *M. monophyllos* had fungal hyphae on the surface (Figure 7A,D), and sometimes other fungal structures, e.g., spores (Figure 7A–F). In a few cases, the authors observed fungal hyphae inside the flower petals (Figure 7B,C,E). During the floral lifespan, the network of fungal hyphae thickened, although there were differences between individual flowers.

### 4.2. Volatile Composition

Living flowers of *M. monophyllos* emitted a complex mixture of volatiles, which were identified with the HS-SPME/GC-MS technique, and 82 volatile compounds were detected. The obtained chromatograms for the two examined individuals overlap (all substances were detected in both samples), although the frequency of some compounds varied significantly between them (Appendix A).

The most frequent group in the *M. monophyllos* volatile composition was the aliphatic compounds (~70–80%), particularly alkanes, alcohols, and aldehydes. The second most frequent group were terpenes (15.5–28.6%), mostly sesquiterpenes (26.9–12%) (Appendix A). n-Heneicosane was the only compound that exceeded 10% of TIC in both samples. Other relatively frequent compounds (>1% of TIC) included: sesquiterpenes: β-elemene, cyperene, β-selinene, α-farnesene; aliphatic ketone: 2-methyl-3-heptanone; aliphatic aldehydes: heptanal, octanal, nonanal, and isomers of octadecatrienal; aliphatic alcohols: 2,3-butandiol, 2-methyl-3-heptanol, 1-heptanol, and 1-octanol; and other alkanes: *n*-Pentadecane, *n*-Eicosane, and *n*-Tricosane. The majority of the substances belonging to the minor components did not exceed 1% of TIC or were in trace (<0.05%) amounts (Appendix A).

### 4.3. Flower Visitor Community

Based on video recordings and direct observations, twenty floral visitation events on the flowers of the *M. monophyllos* population located in Wigry National Park were documented. Flowers were visited solely by dipterans, from at least five families (Table 1). The authors did not find any *M. monophyllos* pollinaria on insect bodies, and thus the authors categorized all of them as visitors. Most visits were made by Drosophilidae: *Drosophila transversa* and *Sceptomyza* species (*S. pallida* or *S. graminium*), which also visited the most flowers during a single bout (on average, N_f_ 3.7 for *Sceptomyza* and 4.0 for *D. transversa*, respectively). The Drosophilidae also spend the longest time on a single flower, and the visit duration on the single inflorescence was the longest (Table 1). Insects representing the fungus gnats, from Keroplatidae/Mycetophyilidae, Sciaroidae families, and members of the family Fannidae and Culicidae, were other common visitors that spent a relatively long time on the *M. monophyllos* flowers. The most often-visiting insects landed directly on the flower, on a dorsal sepal and petals, or on the axis of the inflorescence, and then moved onto the flowers. The individuals of Drosophilidae and Mycetophilidae licked the entire surface of the labellum, but they were clearly interested in the inner, smooth part of the labellum, just behind the column. Only in one case of the *Sceptomyza* species the oviposition-like behaviour was observed. Culicidae, Sciaridae members, and *Fannia norvegica* ‘inspect’ the inner surface of the labellum, although their visits to flowers were much less intense and they much more frequently moved outside the flowers. *D. transversa* spent a long time licking the labellum margin and spent less time looking inside the labellum.

## 5. Discussion

The *Malaxis monophyllos* flower visitors and their behaviour, together with analyses of flower structure and volatile composition, provided essential information on its pollination system. Despite proportionally high pollinia removal [34] and annually observed fruiting, the authors do not detect pollinators in the study population of *M. monophyllos* during presented studies or during the 13 years of this population monitoring. Such a situation is not uncommon among orchids, for many species pollinators have never been observed or were noted sporadically [57,58,59]. Nevertheless, even if the authors accept that only the subset of observed visiting insects acts as pollinators, these observations with *M. monophyllos* flower characters reflect a Diptera-specific pollination system. The flower visitors were, as mentioned, solely dipterans, which is consistent with earlier investigations that have identified both visiting insects and *M. monophyllos* pollinators [31,37]. Despite differences in species composition, both the previous and present observations encompassed two functional groups of potential pollinators. These groups include nectar-feeding insects (Culicidae), as well as species that are attracted to feed or oviposit on fungus and decaying plants (also related to microbe-feeding) (Drosophilidae, Keroplatidae/Mycetophillidae, Sciaroidae, Fanniidae) [60,61,62,63]. This means that the pollination system and floral signals that work in *M. monophyllos* should be considered multifaceted. These kinds of complex systems might be preferred in highly variable environments, where the abundance and distribution of pollinators fluctuate temporally and spatially, and where the plants occur in a great dispersion, like it is in the studied *M. monophyllos* population [11,64]. In any conditions, plants had to find a trade-off between specific pollinators with variable visitation rates and fruit set and more generalist pollinators that may be present in larger abundance. Thus, they may guarantee stable fruiting, but they may also lose pollen more frequently. Our study indicated that this second evolutionary path is most probable for *M. monophyllos* and that, except for specific pollination events, incidental pollination and pollen loss caused by various groups of dipterans that export pollen by chance could happen. Pollen loss seems to be frequent in this species, which is suggested by high pollen removal (90–100%) (extremely adhesive viscidium in the optimum of anthesis—authors observation) and a relatively small and highly varied fruiting level (0.5–40%) [34,35]. Among the insects that the authors observed on *M. monophyllos* flowers, only Keroplatidae/Mycetophillidae and Sciaroidae seem to meet the conditions of successful pollinators due to adequate size and respectively short mouthparts, along with sitting on the flower in the right direction (to remove the pollinaria).

### 5.1. Multifactorial Volatile Composition

In the case of dipterans, fragrance plays an important role in the whole life cycle. They search for food sources, brood sites, or mates, guided mainly by sense of smell [39,40,65]. Thus, floral fragrances may show chemical similarities to compounds involved in insect chemical-communication systems [66]. The *M. monophyllos* pollination strategy seems to focus on luring Diptera, and thus the volatiles might be of particular importance at this point, even more so when the flowers are inconspicuous, greenish, grow in dense vegetation, and their visibility is low. The *M. monophyllos* flower structure analyses revealed a huge aggregation of osmophores in the bent edges of the labellum, as well as three-cell trichomes scattered over the whole corolla, which may be indirect evidence of fragrance emission quality. Ultrastructural TEM analysis confirmed these findings through the presence of the dense cytoplasm contents, i.e., numerous mitochondria, plastids with plastoglobuli and dictyosomes, which are likewise associated with high metabolic cell activity during any secretion [50,67,68]. Furthermore, the extremely strongly folded cuticle of most labellum cells (that may increase the evaporation surface), with a considerable amount of residue of secreted substances, also points to labellum intense secretory function (both volatiles but also nutritional). Finally, highly sensitive techniques for detecting and analysing plant volatiles (HS-SPME/GC-MS) identified several dozen volatile components emitted by *M. monophyllos* flowers. Likely, the production of such a volatile variable set by a single plant species is a more generalist strategy to attract enough pollinators and exploit different aspects of dipteran pollinators’ behaviour [69]. Most compounds were detected in very small amounts, but they are not necessarily less important in the pollination system [19].

The observations of floral visitors showed that *M. monophyllos* flowers are visited exclusively by a few groups of Diptera, probably attracted by the mushroom-like scent that came from the presence of fatty acid-derived compounds: 1-octen-3-ol, 1-octanol, and other eight-carbon compounds, selected monoterpenes (β-pinene), as well as of alcohols, ketones, and aldehydes [70,71,72,73,74,75]. In general, flies show pronounced behavioural sensitivity to aliphatic alcohols and aldehydes [76], which make up almost 50% of the total *M. monophyllos* fragrance. Thus, the sapromyiophily is one of the plausible ways in which semiochemicals work in the *M. monophyllos* pollination system. The majority of Drosophilidae species, as well as Keroplatidae/Mycethophilidae, Sciaroidae, and Fanniidae (all observed on *M. monophyllos* flowers) have saprophagous larvae and adults feeding on fungi, but also on yeast and bacteria living on decomposing plant material [41,59,61,74,77,78,79]. However, mushroom-visiting flies may also respond to other rewards, such as shelter or eking sites [80,81]. Orchids with flowers visited by this group of flies most often have a pollination system based on brood site deception, which is well documented, for example, in the genus *Dracula* [82,83,84]. Other orchids, i.e., *Corybas cheesemanii*, or *Pleurothallis marthae*, combine brood site (fungus) deception with other attractants (nectar resources) to lure ovipositing females of fungus gnats [85,86]. Nevertheless, except for one observation of *Scaptomyza* species, the authors did not observe any oviposition behaviour on *M. monophyllos* flowers, nor did the authors observe the laid eggs of any insects. Regardless of the functions performed, fungus or decaying plant substrates are highly ephemeral, and insects that use them have a variable probability of finding suitable sites for laying their eggs or finding food resources. Thus, plants with flowers that mimic such sites tend to have pollination systems that are relatively generalist at the insect species level [40,41,43,82]. Mycethophilidae and Sciaroidae were described in literature as the most important fungus gnat pollinators, although they may likewise represent various functional groups [39,87]. A nectar reward with brood site deception is the pollination system that was described for the orchid *Pleurothallis marthae* [86]. The second group is rewarded with nectar, as described for *Listera cordata* [88]. The third group consists of species that pollinate flowers also by deceit, specifically sexual deception similar to the orchids *Lepanthes* or *Pterostylis* [89,90,91].

The specific blends of terpenes, aliphatic alkanes, aldehydes, and alcohols identified in the *M. monophyllos* volatiles were previously described as pheromones, be they sexual or aggregation, implicated in deceptive, mainly sexually deceptive, pollination systems [43]. One of the best documented sexual deception strategies in the orchid is represented by *Ophrys* and males of solitary bees [28,92,93]. Active compounds of *Ophrys* fragrance composition, e.g., alkanes, including heneicosane, docosane, and tricosane, are described as sex mimicry pheromones. These are also present in *M. monophyllos* fragrance. In total, they account for about 20% of all fragrance fractions (Appendix A), which may indicate the important role of these compounds in its pollination system. In addition, tricosane and its derivatives play a role in regulating some *Drosophila* mating and courtship behaviours [94]. The fatty acid-derived compounds are also essential volatiles in plant pollination by mosquitoes, relatively frequent visitors of *M. monophyllos* flowers. The octanal and the nonanal-rich scent of *Platanthera obtusata* and floral odourants of *Tanacetum vulgare* in which 1-octen-3-ol and nonanal dominate attract *Aedes* and *Culex* mosquitoes [95,96]. Besides aromatic compounds, within *M. monophyllos* volatiles, the authors detected a large number of terpene blends (>10–20%, e.g., β-elemene and (*Z*,*E*) and (*E*,*E*)-α-farnesene). They were also described as active ingredients, including pheromones enticing butterflies, termites, and aphids [22,97,98]. Our research showed a complex scent bouquet of *M. monophyllos* flowers that requires further physiological and behavioural analysis to elucidate the influence of the particular fragrance components on putative pollinators.

### 5.2. Structure and Colour of M. monophyllos Flowers

The combination of visual and olfactory signals often has a greater influence on pollinator behaviour than either of these cues alone. The *M. monophyllos* flower structure analysis provides additional insight into its pollination ecology. The rewarding of pollinators has rarely been addressed in the literature about genus *Malaxis* [29]. Claessens, and Kleynen [31] described liquid droplets on the labellum, although it was not clear if it was a nutritional liquid. The Keroplatidae/Mycetophilidae and Culicidae species visiting *M. monophyllos* flowers tasted this inner part of the labellum. Further analyses using scanning microscopy revealed the presence of a polished surface in this area of the labellum not indicted by SEM procedure and giving the impression of liquid (macroscopic observations). That allowed the authors to indicate a large amount of secretion, probably with a waxy character that makes the surface smooth and suitable for licking. Moreover, Sudan black B staining revealed a large accumulation of lipid substances in the epidermal and subepidermal cells of this part of the labellum. For this reason, pores in the cuticular surface and the remnants of secretions on this polished surface may display food-rewarding attributes for pollinators. Moreover, this area’s location behind the gynostemium could also act as a visual signal that may guide insects to look for the nectar inside the labellum, promoting effective pollination.

Our observations of some visiting insects on *M. monophyllos* flowers suggest the presence of a reward also on the labellum margin. This part of the flower was intensively tasted, especially by Drosophilidae *Drosophila transverse* and *Scaptomyza* species. Furthermore, the ultrastructure analyses of cells revealed traces of metabolic activity with residues of secreted substances in almost all labellum parts, which may be one of arguments for interpreting them as nutritional for pollinators. Nevertheless, other nutritional items may also be considered here. The differential staining and scanning microscopy revealed the presence of fungal hyphae (and other fungus structures). Some grow into the cells through holes in cell walls made during raphide ejections. The degree of fungus hyphae coverage depends on flower age, but the authors found single hyphae even on flowers at the beginning of anthesis. In the case of the orchid *Dracula felix*, authors found that yeast and mould species inhabited the labellum and other floral structures, which may play a role in pollination efficiency by the attraction and retention of mushroom-feeding flies [99]. Fungi may be introduced to the flowers by flies or airborne means, and their contribution in the *M. monophyllos* pollination system is certainly worth further exploration.

Moreover, the behaviour of various dipterans is further enhanced by visual stimuli, for example, ultraviolet floral reflections that often guide nectar-foraging insects [85,100]. Also, fungus gnat and mosquito visits occurred more frequently at dusk [87]. Because of the complex visual system operating in the eyes of most flies, they have tetrachromatic colour vision, with sensitivity from ultraviolet (UV) to yellow wavelengths [101] that is of great importance in low light conditions at dusk. The authors found UV luminescence in the marginal part of the *M. monophyllos* labellum to be responsible for the UV reflectance pattern of flowers. Such visual signals are confirmed as being more conspicuous for insects than flowers with uniform reflectance [100]. For the small dipterans, for example, that use mushrooms as food or brood sites, the luminescence of mushrooms is a valid decoy [102]. Biological luminescence results from various mechanisms, including crystalline inclusion [103]. The authors found the source of such luminescence in the peculiarly bulged structures of the *M. monophyllos* labellum margin, where there was an exceptional amount of idioblasts with the crystal raphides, which may be responsible for this visual signal.

Further analyses of idioblast structure revealed that they are mostly subepidermal elements, but some walls of these cells are in contact with the external environment. The raphides in plants are usually found in crystal idioblasts in the parenchymatous tissue [104,105], but also in subepidermal cells [106,107]. Raphides from the margin of the *M. monophyllos* labellum were sometimes ejected from these cells, which can be related to changes in the cell wall structure and the mechanical stimulus, although the exact mechanism requires further investigation. In general, raphides in the flowers of the Malaxidinae subtribe are common, and their location (around nectar concavities and gynostemium) is putatively part of nectar guides visible in UV light. In the case of Malaxidinae, but also found in the genera of *Stelis* and *Corybas*, it has been previously proposed that extracellular oxalate crystals could facilitate pollination by providing a visual signal, including nectar illusion, or a scent (so-called green volatiles) that might be interesting for some groups of insects [108,109,110,111].

## 6. Conclusions

Flowers attract pollinators by their form and odour, acting as a sensor trap for insects searching for food and oviposition sites or mating partners [39]. Even if the authors accept that only a subset of insects observed during this study acts as *M. monophyllos* pollinators, both visual and olfactory signals appear to be dedicated to dipterans, and the synergistic effect of all attractants seems to be necessary to elicit visiting and pollination of *M. monophyllus* flowers. Considering the variety of emitted volatile compounds and features of flowers’ anatomical structure, in the context of visiting insect behaviour, the authors can make preliminary assumptions about a few signalling modalities that are attractive to guilds of *M. monophyllos* pollinators. Due to infrequent visits of pollinators and low to moderate fruit set in *M. monophyllos* populations, further supported by the composition of scent bouquet, the authors can assume that it display some attributes of deceptive taxa and employs both brood site and food decoy to lure some of its pollinators. However, the entire *M. monophyllos* labellum exhibits metabolic and secretory activity that can related to (1) emission of volatiles and (2) to visual attractants, but also (3) the nutritional reward, which should be also considered. Thus, further detailed experiments are needed to demonstrate the synergistic effect of these multimodal signals and clearly define whether *M. monophyllos* flowers offer a reward to its pollinators, both searching for nutritional secretions and fungus/microbe feeding. This work reveals a few new issues in *M. monophyllos* pollination biology that creates possibilities for future in-depth insights in several scientific areas.

## Figures and Tables

**Figure 1 biology-11-00640-f001:**
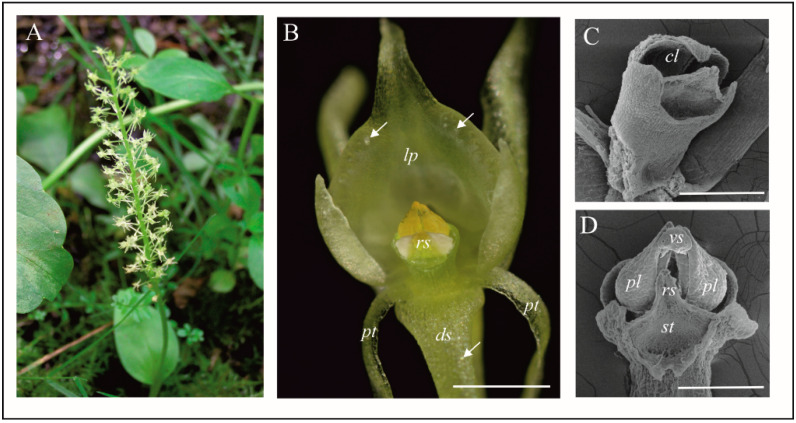
General plant and flower morphology of *Malaxis monophyllos*. (**A**) flowering plant in the natural habitat of Wigry National Park; (**B**) flower in optimum of anthesis, arrows—indicate raphides in the cells of the labellum; (**C**) gynostemium after removing the pollinaria; (**D**) gynostemium with pollinaria (SEM). Scale bars: 500, 200, 200 µm. *cl*—clinandrium, *ds*—dorsal sepal, *lp*—labellum, *pl*—pollinium, *pt*—petal, *rs*—rostellum, *st*—stigma, *vs*—viscidium.

**Figure 2 biology-11-00640-f002:**
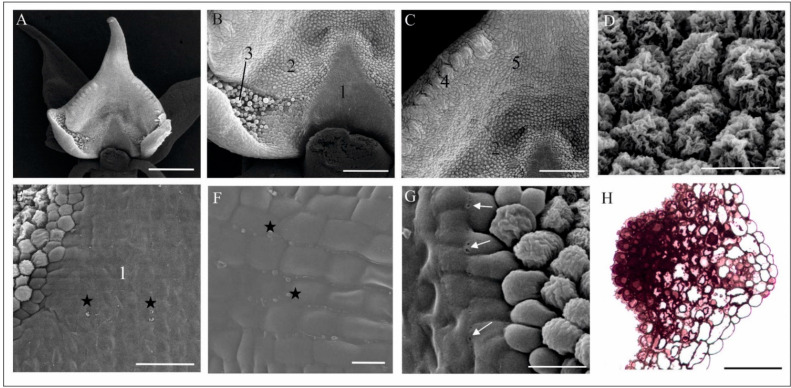
The micromorphology and histochemistry of the *M. monophyllos* labellum (SEM, LM). (**A**,**B**) Labellum—a frontal view with the folded margins and epidermal cells differentiated in terms of the cuticle structure and secretory function: (1) a triangle-shaped area with a smooth surface located behind the gynostemium; (2) cells of a strongly undulated cuticle; (3) glandular trichomes in the lateral, folded margins of the labellum; (**C**) upper parts of the labellum with distinct areas: (4) the bulging cells along the edges of the labellum, (5) the cells of the central part of the labellum; (**D**) details of the inner part of the labellum (2), made of cells with the strongly undulated cuticle; (**E**–**G**) details of triangle-shaped area (1); (**E**,**F**) smooth surface (1) formed by thick cuticular coat covering the cells, with residues of secreted substances (asterisk); (**G**) pores (arrows) on the edge of the waxy coat; (**H**) lipid concentration in epidermal and subepidermal cells of the inner part of the labellum (1) (semisection treated with SBB). Scale bars: 500, 200, 200, 20, 50, 10, 20, 50 µm.

**Figure 3 biology-11-00640-f003:**
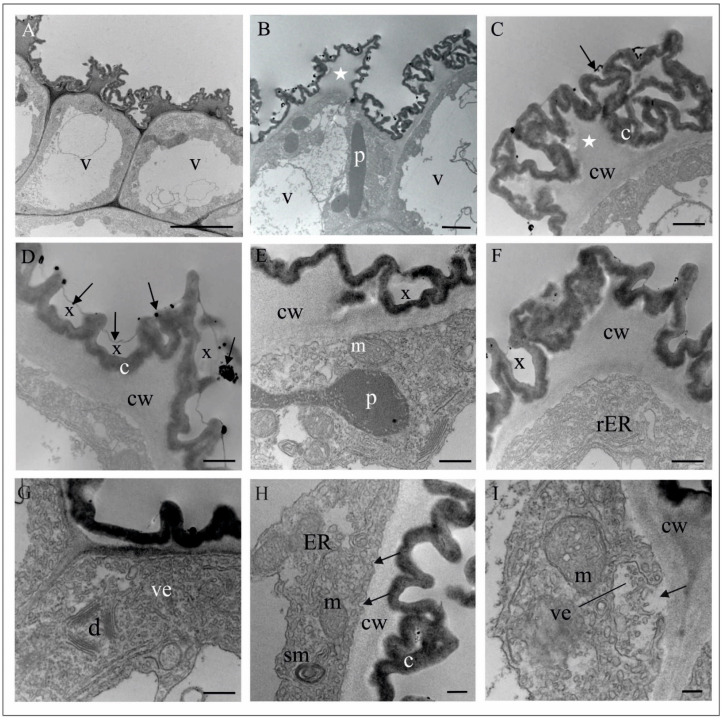
Ultrastructure of the inner surface of the *M. monophyllos* labellum: cells with the strongly undulated surface (shown in SEM: Figure 2D). (**A**–**C**) Strongly vacuolated epidermal cells; the outer wall with irregular outgrowths (stars), covered with folded cuticle; (**C**–**F**) The numerous crypts of the folded cuticle with homogeneous material (x) secreted from the cells with an osmophilic sediments and granules on the surface (arrows); (**A**–**I**) the dense parietal cytoplasm containing plastids with plastoglobuli, mitochondria, dictyosomes, and extended membrane system (profiles of rough and smooth reticulum, numerous vesicles and myelin-like structures). Single vesicles fusing with plasma membranes near the cell wall (**H**, arrows); whole vesicles are released beyond the protoplast ((**I**), arrow)). Scale bars: 2 µm, 2 µm, 1 µm, 500, 500, 500, 500, 500, 500, 500, 200, 200 nm. *c*—cuticle, *cw*—cell wall, *d*—dictyosome, *m*—mitochondrion, *p*—plastid, *rER*—rough endoplasmic reticulum, *ER*—endoplasmic reticulum, *v*—vacuole, *ms*—myelin-like structures, and *ve*—vesicles.

**Figure 4 biology-11-00640-f004:**
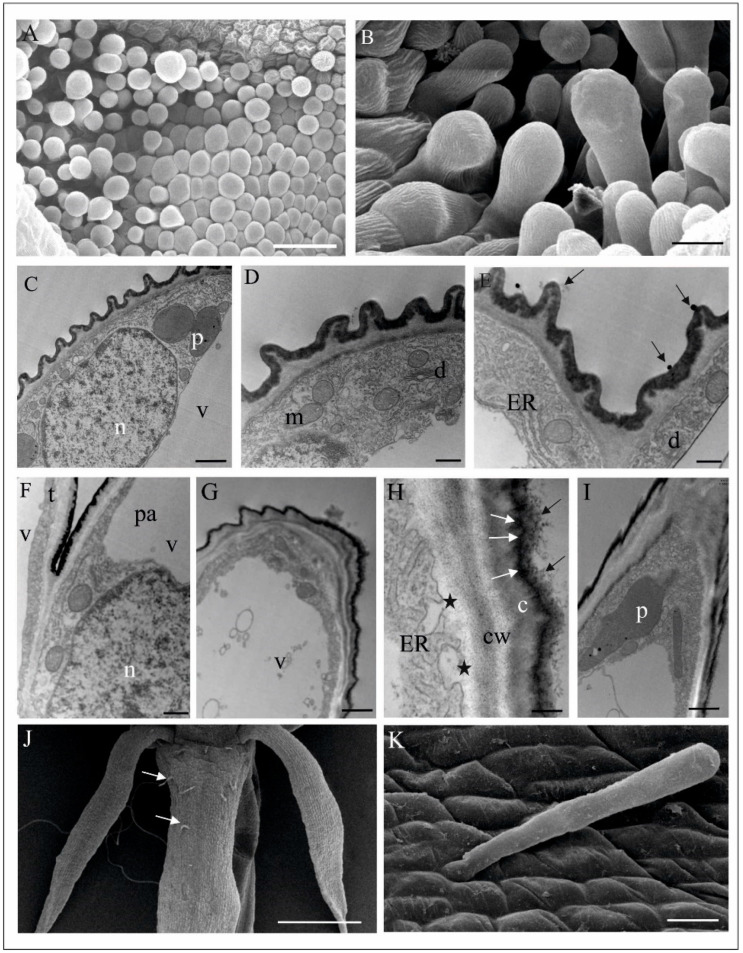
Papillae and trichomes in the folded margins of *M. monophyllos* labellum (A-I) (Figure 2B—marked with 3). (**A**) Papillae and trichomes—top view (SEM); (**B**) trichomes with the cuticular striations—side view (SEM); (**C**–**E**) ultrastructure of the lenticular-convex papillae (TEM); outer cell wall with appendages covered with cuticle and remnants of secretive materials (arrows), the cytoplasm containing plastids with plastoglobuli, mitochondria, dictiosomes, polyribosomes on the surface of an ER cisterna and numerous vescicles; (**F**–**I**) ultrastructure of the trichomes (TEM): fragment of the lower part of the two trichomes (**F**): bicellular (t) and unicellular, finger-like (pa); the cells of both trichomes highly vacuolated; (**G**–**I**) details of bicellular trichome: (**G**) part of the one-cell stalk, under the head cell; (**H**,**I**) Part of the longitudinal section through the strongly vacuolated head cell of trichomome; the cell wall with thickened irregularly, with protrusions; huge amounts of secreted material (black arrows) on the cuticle surface and streaks of such material transported across the cell wall (white arrows); an extensive system of ER cisternae in the cytoplasm; the vesicles incorporated into the plasmolemma release their contents to the outside (asterisk); (**J**,**K**) 3-cells trichomes on a dorsal sepal (white arrows) (SEM). Scale bars: 50 µm, 20 µm, 1 µm, 500 nm, 500 nm, 500 nm, 1 µm, 200 nm, 1 µm, 500 µm, 20 µm. *c*—cuticle, *cw*—cell wall, *d*—dictiosome, *ER*—endoplasmic reticulum, *m*—mitochondrion, *n*—nucleus, *p*—plastid, *v*—vacuole.

**Figure 5 biology-11-00640-f005:**
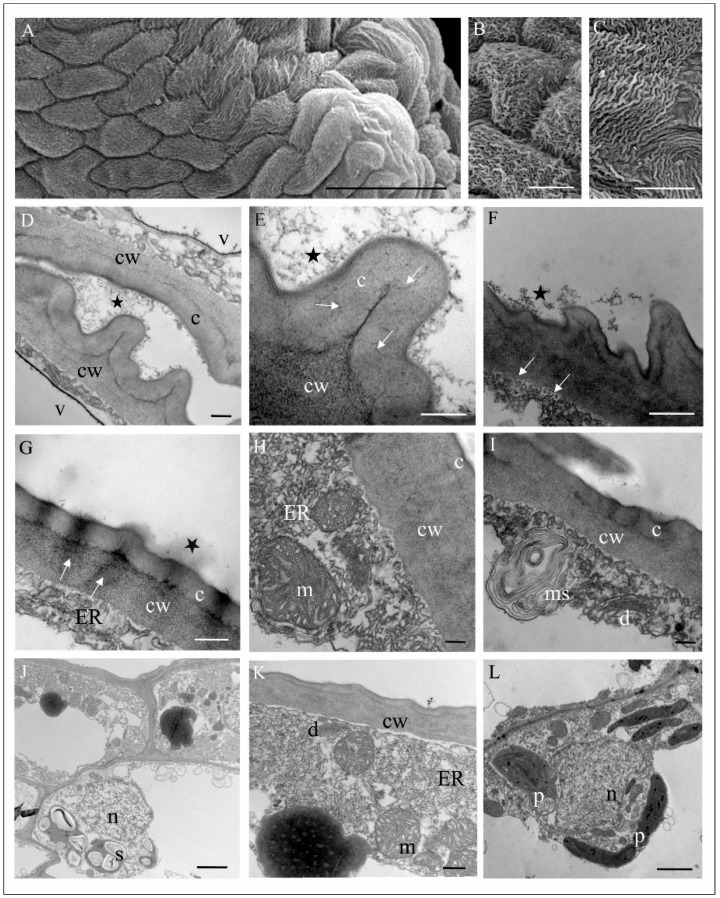
Structure of the middle and marginal part of the *M. monophyllos* labellum (Figure 2B—marked with 4 and 5). (**A**–**C**) Epidermal cells with different pattern of cuticular striations (SEM); (**D**,**E**) epidermal cells with the cuticle and the micro-channels (arrows); residues of secreted, grainy-flocculent substance on the cuticle surface (asterisk); (**F**) folded plasma membrane with the vesicles in periplasmic space (white arrows); (**G**) osmophilic streaks of material transported across the cell wall (white arrows) with extruded homogeneous substances on the cuticle (asterisk); (**D**–**I**) the outer walls of the epidermis of varying thickness, smooth or with outgrowths of the cell wall with different size and shape; parietal cytoplasm of epidermal cells rich in extended profiles of ER, dictyosomes, vesicles and myelin-like structures; strongly folded plasma membrane; (**J**–**L**) epidermis and subepidermis; the latter contains plastids with starch grains (**J**) and plastids with plastoglobuli (**L**). Scale bars: 50 µm, 10 µm, 10 µm, 200 nm, 100 nm, 200 nm, 2 µm, 200 nm, 200 nm, 2 µm, 200 nm, 2 µm. *c*—cuticle, *cw*—cell wall, *d*—dictiosome, *ER*—endoplasmic reticulum, *m*—mitochondrion, *ms*—myelin-like structures, *n*—nucleus, *p*—plastid, *s*—starch grains.

**Figure 6 biology-11-00640-f006:**
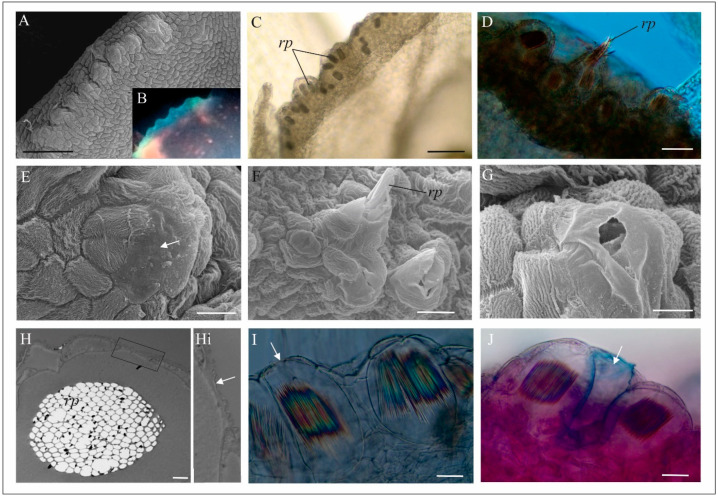
The cells with raphides in the *M. monophyllos* labellum. (**A**) The bulged structures on the labellum margin (SEM); (**B**) Luminescence of the labellum margin in the UV light (330–380 wavelengths) (LM); (**C**) idioblast with the raphides (fixed cells without staining) (LM); (**D**) raphides after ejection from the idioblasts (LM, DIC); (**E**–**G**) The bulged structures on the labellum margin-details (SEM); cells with striped cuticle and a single cells with smooth cuticle (arrow); visible ejected raphides (**F**) and perforation (**G**) in the thin, smooth cuticle; (**H**) bundle of the raphides in the idioblast; (**Hi**) inset—fragment of the idioblast cell wall borders with the external environment (TEM); (**I**,**J**) Idioblasts with the raphides in the subepidermal cells; the fragment of the cell wall borders with the external environment (arrow); (**I**) and idioblast without raphides and with torn cell wall (**J**) (LM, DIC). Scale bars: 100, 100, 20, 20, 20, 20, 10, 20, 20 µm. *e*—epidermis, *rp*—raphides.

**Figure 7 biology-11-00640-f007:**
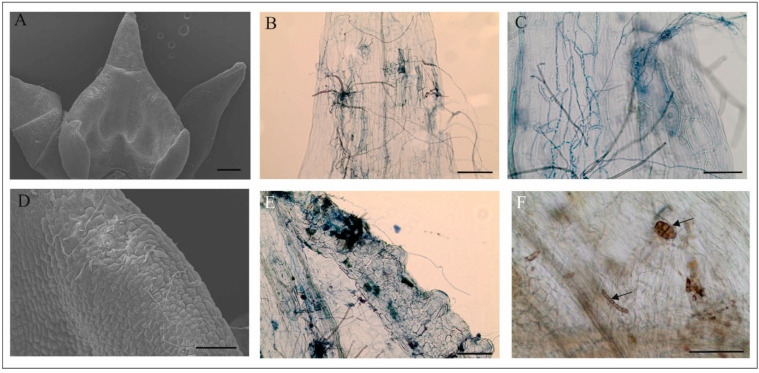
*M. monophyllos* labellum with a wide range of fungi both on the surface and inside tissues. (**A**–**F**) Both fungal hyphae and spores (**F**, arrow) are visible. Scale bars: 200, 20, 50, 50, 50, 50 µm.

**Table 1 biology-11-00640-t001:** Insects visiting *Malaxis monophyllos* flowers in the studied population with quantitative components of insect activity.

Order	Family	Species/Genus	N	N_f_	T_f_ (s)
					(Mean ± SD)
Diptera	Drosophilidae	*Drosophila transversa* Fallén, 1823	4	4.0 *	66.8 (±122.2)
		*Scaptomyza pallida* (Zetterstedt, 1847) or *S. graminum* (Fallén, 1823)	3	3.7 *	150.3 (±69.43)
	Sciaroidae	*Sciara* sp.	2	2.0 *	52.5 (±67.18)
	Mycetophilidae	Unidentified	2	3.0 *	52.4 (±28.85)
	Mycetophilidae or Keroplatidae	Unidentified	1	1	5
	Fanniidae	*Fannia* cf. *norvegica* Ringdahl, 1934, ♀	1	3	20
		*Fannia* sp., ♀	1	1	>240
	Culicidae	Unidentified ♀, taxon 1	4	1.2 *	10.5 (8.75)
		Unidentified ♀, taxon 2	2	2.0 *	12.2 (12.97)

N—number of recorded/observed insects; N_f_—number of flowers per inflorescence visited during a single bout (* mean); T_f_—visit duration per flower (seconds).

## Data Availability

All data generated or analysed during this study are included in this published article (and its Appendix A). Exceptions are the video recording, that will be available from the corresponding author on reasonable request.

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
