# Peer review of "The Floral Signals of the Inconspicuous Orchid Malaxis monophyllos: How to Lure Small Pollinators in an Abundant Environment"

_biology, 2022, doi:10.3390/biology11050640_

Round 1

Reviewer 1 Report

  • The general impression is that the authors do not bring much of a new results with this paper. Despite of that the text is too long, contains too many details, the authors used too many references, and wrote too long discussion. I believe that some parts of the text should be compressed, number of references and figures reduced, general statements (like on pollination) deleted.
  • Page 3 - In section Study species and population authors stated that plants were collected in 2014 (43 shoots), 2017 (155) and 2019 (145), with 30% of flowering shoots. In the next chapter Plant material authors stated that flowers were collected in 2017. Which one is true? The exact number of analyzed flowers and sections should be given (LM, SEM, TEM).
  • page 5, reference Jermakowicz et al 2015 should be changed to (35)
  • The authors give so many unnecessary details in description of labellar epidermis anatomy. Some general issues, like cell organelles of the secretory cells, are already known and there is no need to write about them in detail.
  • page 6, paragraph 1 – Please check the sentence structure “The more that the numerous crypts…” and “Additionally indication are…”
  • page 6, paragraph 3 – it says “Fig. 2 marked as 5” here. If you look under the Fig 5. on which it refers to at the page 11, it says “Fig 2 marked as 4” – which one is true?
  • page 6, paragraph 4 – None of the figures is marked as 6B
  • Figure 1 – lb and ls are not marked at the figure. What is lp on the figure?
  • Legend of the Fig 3, second line – there is no “I” in Fig 2 (“shown in SEM: Fig 2 G-I”)
  • Figure 3, 4, 5 and 6– are all the parts of these figures really necessary? They seem repetitive and too many.
  • Legend of the Fig 5 – instead of (D-I) it should be (H-I)
  • Figure 6 – the 6B is missing
  • page 15, paragraph 1 – Please check the sentence structure “A nectar reward…”
  • references 18-19 are not on the list

Author Response

Response to Reviewer 1

The authors thank to Reviewer comments and suggestions. We have provided our responses to the reviewers’ comments and made amendments in the text of the manuscript.

Below we have written the detailed description of our responses to all comments. Responses are reported in blue font. We have also asked a native speaker to correct the language in final version of the manuscript.

We hope it is currently suitable to be publish in Biology.

Kind regards,

The authors

Comment: The general impression is that the authors do not bring much of a new results with this paper. Despite of that the text is too long, contains too many details, the authors used too many references, and wrote too long discussion. I believe that some parts of the text should be compressed, number of references and figures reduced, general statements (like on pollination) deleted.

Response: We could not quite agree. The pollination biology of M. monophyllos has never been study in details so far. Except the Classens and Kleynen (2011) that catch the M. monophyllos pollinators in Alpine populations and gives some general statements about pollination biology. Although the information of M. monophyllos pollination biology in many points were vague. Our study give some completely new insight in the pollination biology of this species, for example regarding flower scent contents, secretory activity of the flower tissues, presence of fungal structures in the flowers, or mechanisms of raphides rejections. The study concerned one species of orchid, but many finding seems to be important in resolving Diptera specific pollination systems.

According the reviewer suggestion we reduced some part of the manuscript, specifically number of photos in Figure 3, and number of references. We also rephrase the statements regarding pollination biology.     

Comment: Page 3 - In section Study species and population authors stated that plants were collected in 2014 (43 shoots), 2017 (155) and 2019 (145), with 30% of flowering shoots. In the next chapter Plant material authors stated that flowers were collected in 2017. Which one is true? The exact number of analyzed flowers and sections should be given (LM, SEM, TEM).

Response: Our analyses were quite complex and consisted with visiting insects observations that were initiated in 2014 and repeated in 2019. Plant material collection were also conducted twice, first in 2017 and in 2019 for SEM and TEM analyses, and finally scent analyses which were done in 2019. The number of shoots (43,155,145) refers to the size of the population in a given year, which highly varied between years. In general flowers from 5 M. monophyllos shoots were collected. From each shoot 5 flowers that further were analyzed anatomically.  

Comment: page 5, reference Jermakowicz et al 2015 should be changed to (35)

Response: It have been done.

Comment: The authors give so many unnecessary details in description of labellar epidermis anatomy. Some general issues, like cell organelles of the secretory cells, are already known and there is no need to write about them in detail.

Response: The secretory activity of M. monophyllos flowers tissues haven’t been reported so far, thus gives some details and evidences of such activity is justified and necessary in this case, the more that it opened the new scientific areas in investigation of pollination biology of this species.

Comment: page 6, paragraph 1 – Please check the sentence structure “The more that the numerous crypts…” and “Additionally indication are…”

Response: We have rephrased this sentence.

Comment: page 6, paragraph 3 – it says “Fig. 2 marked as 5” here. If you look under the Fig 5. on which it refers to at the page 11, it says “Fig 2 marked as 4” – which one is true?

Response: We have corrected it. Figure 5 applies the area of labellum marked with 4 and 5 in the Figure 2.

Comment: page 6, paragraph 4 – None of the figures is marked as 6B

Response: Figure 6 have the photo B.

Comment: Figure 1 – lb and ls are not marked at the figure. What is lp on the figure?

Response: The markings were changed, we improved it.

Comment: Legend of the Fig 3, second line – there is no “I” in Fig 2 (“shown in SEM: Fig 2 G-I”)

Response: We improved it according to markings on the figure.

Comment: Figure 3, 4, 5 and 6– are all the parts of these figures really necessary? They seem repetitive and too many.

Response: All the figures show micro and ultrastructure of the different parts of the labellum, and are the evidences of secretory activity of cells in almost all parts of M. monophyllos labellum – a new and very important element of M. monophyllos pollination biology. In the figure 6 we showed the location and mechanisms of raphids ejection from the cells – also new and intriguing mechanism previously do not described in this species. In general our study raised some new questions in pollination biology of this orchid species and showed that only complex, multifaceted approaches could gave the true picture of pollination biology of any plant species.

Comment: Legend of the Fig 5 – instead of (D-I) it should be (H-I)

Response: The differences in the cell walls of this part of the labellum have been showed on 5 D-I photos.

Comment: Figure 6 – the 6B is missing

Response: Figure 6 have the photo B.

Comment: page 15, paragraph 1 – Please check the sentence structure “A nectar reward…”

Response: We have rephrased this sentence.

Comment: references 18-19 are not on the list

Response: We include it into the list.

Reviewer 2 Report

Jermakowicz et al. provided micro- and morphological characters of flowers of Malaxis monophyllos. Floral scents were analyzed. These characters related to pollination syndromes.  They also obeserved the visitors by video and direct observation. I have several comments.

(1) The genus Malaxis was redelimited. Please cited the reference about generic delimitation.

(2) In total, twenty floral visitation events on the flowers of the M. monophyllos were documented. However, no pollinator has been confirmed. Therefore, most of discussion about pollination are postulation.

(3) It is very importan to state that how many hours on direct observation? How many plants were observed?

(4)If possible, please state that whether the traced nectar could be collected and analyzed for their elements?

(5) Malaxis monopllyos is widespread from Asia to Europe. How about the pollination systems across such wide range?

Author Response

Response to Reviewer 2

The authors thank to Reviewer for comments. Below we have written the detailed description of our responses to all comments. Responses are reported in blue font. We have also asked a native speaker to correct the language in final version of the manuscript. We hope it is currently suitable to be publish in Biology.

Kind regards,

The authors

Jermakowicz et al. provided micro- and morphological characters of flowers of Malaxis monophyllos. Floral scents were analyzed. These characters related to pollination syndromes.  They also obeserved the visitors by video and direct observation. I have several comments.

(1) The genus Malaxis was redelimited. Please cited the reference about generic delimitation.

Response: Unfortunately I do not know this paper, thus I couldn’t refer on it. 

(2) In total, twenty floral visitation events on the flowers of the M. monophyllos were documented. However, no pollinator has been confirmed. Therefore, most of discussion about pollination are postulation.

Response: We are fully agree with the reviewer, that pollinators observations are the most reliable, although in many species of orchids pollinators with pollinia have never been observed, which does not mean that nothing can be said about their pollination systems. Despite we couldn’t managed to catch M. monophyllos pollinators, we have observed the visiting insects which they were exclusively Diptera. We think that our multifaceted study covering analyses of flowers structure, scent biochemistry and insects behavior are very valuable and make the significant contribution in understanding the pollination biology of this species.

(3) It is very importan to state that how many hours on direct observation? How many plants were observed?

Response: It was about 30 hours in 2014 (in sum 9 inflorescence) and 42 hours in 2019 (in sum 15 inflorescence). All this information are included in the methods.

(4)If possible, please state that whether the traced nectar could be collected and analyzed for their elements?

Response: These are micro-quantities thus collecting is rather impossible, although histochemical analyses, with specific staining of carbohydrates, proteins and other nutritional substances could be possible.  

(5) Malaxis monopllyos is widespread from Asia to Europe. How about the pollination systems across such wide range?

Response: There are observations of pollinators in alpine population of M. monophyllos done by Classens and Kleynen (2011) that catch the pollinators and gives some general statements about its pollination biology.

Reviewer 3 Report

The manuscript and the author's work is clear. I think that the result needs to be checked for what concerned the English grammar. 
All pictures are well done. 

Author Response

Response to Reviewer 3

The manuscript and the author's work is clear. I think that the result needs to be checked for what concerned the English grammar. 
All pictures are well done. 

The authors thank to Reviewer for appreciating our work. We have also asked a native speaker to correct the language in final version of the manuscript.

We hope it is currently suitable to be publish in Biology.

Kind regards,

The authors

Round 2

Reviewer 1 Report

The authors did not make any changes (other than technical, like figure numbers and letters) in the manuscript according to my suggestions. Therefore, I leave the final decision on this manuscript to the Editor.

Reviewer 2 Report

The revision has improved greatly, although no more data were included. Authors have tried their best to improve manuscript. I have no more question.